# Response Mechanism of Leaf Area Index and Main Nutrient Content in Mangrove Supported by Hyperspectral Data

Xiaohua Chen [1,†], Yuechao Yang [2,†], Donghui Zhang [3,*], Xusheng Li [2], Yu Gao [4], Lifu Zhang [3], Daming Wang [5], Jianhua Wang [3], Jin Wang [3] and Jin Huang [6]

1 Hainan Academy of Forestry (Hainan Academy of Mangrove), Haikou 570100, China
2 National Key Laboratory of Remote Sensing Information and Imagery Analyzing Technology, Beijing Research Institute of Uranium Geology, Beijing 100029, China
3 Aerospace Information Research Institute, Chinese Academy of Sciences, Beijing 100094, China
4 Tianjin Key Laboratory of Hyperspectral Technology, Tianjin Progoo Information Technology Co., Ltd., Tianjin 300392, China
5 Tianjin Centre of Geological Survey, China Geological Survey, Tianjin 300170, China
6 School of Geographical Sciences, China West Normal University, Nanchong 637009, China
* Correspondence: zhangdonghui@aircas.ac.cn
† These authors contributed equally to this work.

**Abstract:** Mangrove is the key vegetation in the transitional zone between land and sea, and its health assessment can indicate the deep-level ecological information. The LAI and six key nutrients of mangrove were selected as quantitative evaluation indicators, and the decisive evaluation method of mangrove growth was expected. The mangrove reserve of Dongzhai Port National Nature Reserve in Hainan Province, China, was selected as the study area, with an area of 17.71 km². The study area was divided into adjacent urban areas, aquaculture areas, and agricultural production areas, and key indicators are extracted from satellite hyperspectral data. The extraction process includes spectral data preprocessing, spectral transformation, spectral combination, spectral modeling, and precision inspection. The spatial distribution of LAI and six key nutrient components of mangrove in the study area were obtained. LAI and Chla need to calculate the index after high-order differentiation of the spectrum; MSTR and Chlb need to calculate the envelope after the second-order differential of the spectrum; TN and TP are directly changed by original or exponential spectrum; the spectral transformation method adopted by TK was homogenization after first-order differential. The results of the strength of nutrient content along the three regions show that there was no significant difference in the retrieval index of mangroves in the three regions, and the overall health level of mangroves was consistent. Chla was the key identification component of mangrove growth and health. The contents of nutrient elements with correlation coefficient exceeding 0.80 include MSTR and TK (0.98), Chla and TP (0.96), Chla and TK (0.87), MSTR and Chla (0.86), MSTR and TK (0.83), and MSTR and TP (0.81). The study quantifies the relationship between different LAI and nutrient content of mangrove leaves from the perspectives of water, leaf biology, and chemical elements, which improved our understanding of the relationship between key components during mangrove growth for the first time.

**Keywords:** mangrove; hyperspectral remote sensing; spectral transformation; leaf area index; plant nutrition; remote sensing inversion; correlation analysis

## 1. Introduction

Mangrove has become a sensitive area in response to global climate change due to its coastal distribution, geographical location, and special environment. Extracting vegetation leaf area index from remote sensing observation has always been a difficult and hot issue in the field of quantitative remote sensing [1]. The existence of mangroves also plays a key role in the balance of the global carbon cycle [2,3]. The evaluation of mangrove information

indicators at different spatial scales is realized based on multi-source and multi-scale data such as satellite hyperspectral, ground hyperspectral, and chemical analysis, providing technical support for the establishment of a space–earth integrated monitoring platform that can be shared, compatible, and sustainable [4]. The information of mangrove and non-mangrove areas, different population densities, main species composition, and the impact of human activities can meet the basic requirements of government departments for mangrove mapping and inventory with the combination of multiple remote sensing sensors [5]. The introduction of remote sensing technology into the monitoring and detection of mangroves focuses on mangrove identification, mangrove health assessment, and mangrove physiological and biochemical information extraction [4,6–8].

The first research direction is the accurate extraction of mangrove distribution assisted by remote sensing technology. The traditional method is based on the idea of threshold; a mangrove index is established using Sentinel-2 image to achieve accurate extraction of submerged mangroves in view of the fact that mangroves will be flooded by tidal water, rainy climate, and other adverse factors [9,10]. Additionally, the maximum likelihood classification technology of remote sensing images is introduced to accurately distinguish mangroves [11–13]. With the improvement in data resolution, the satellite data with high spatial resolution can accurately extract the boundary information of mangrove from the texture under the action of a neural network algorithm [14,15]. Additionally, then, a quantitative classification model of mangrove ecosystem degradation was developed using satellite observation data [16–18]. The further application direction is that the carbon storage of mangroves can be roughly estimated using the spatial distribution information calculated by Landsat8, Worldview-2, and ASTER data [19–21]. Then, a mangrove extraction model was established to effectively capture the spatial distribution of mangroves from sparse to dense, with different forms based on the spectral index analysis [22,23]. Remote sensing technology can collect, process, and image the reflected or radiated electromagnetic wave information to detect and identify various objects on the ground, thus accurately extracting the distribution of mangroves. Optical remote sensing images taken in different bands can help to distinguish the distribution range of mangroves, and radar remote sensing technology can also obtain data with different reflectivity because of the high water content of mangroves, which makes them easier to detect.

Secondly, mangrove health assessment based on multi-source remote sensing data has also been extensively explored. Remote sensing methods have been proven to be effective in mapping mangrove species, estimating their biomass, carbon storage, and assessing the range change [24]. Combined with satellite data and ground survey data, it is proven that mangroves have good economic and social value [25]. The precise mangrove map generated by using Sentinel-1 and Sentinel-2 images, combined with Google Earth Engine (GEE), provides a new technology for the evaluation of mangrove ecosystem functions [26–29]. The time series data of satellite sensors have become a necessary means to quantify the changes of mangrove cover at the regional and global levels in order to understand the changes of mangrove growth with time [30,31]. The remote sensing monitoring data in the past 40 years show that the mangrove area changes significantly from year to year in the Guangdong province. Natural factors such as temperature, sea level rise, extreme weather events, and coastline length have a macro impact on the distribution of mangroves [32,33]. The whole process of mangrove death caused by road construction was studied based on the analysis of optics, synthetic aperture radar, UAV image, and topographic survey data [34,35]. The spatial and spectral information obtained through remote sensing technology can reflect the growth status and species composition of mangroves, as well as the impact of habitat changes and other factors on the health of mangroves. Using remote sensing parameters to analyze vegetation cover, relative productivity, leaf area index, and other parameters of mangroves, the health status of the forest can be inferred. Meanwhile, combining multiple sources of information such as ground surveys and meteorological observations can improve the accuracy of mangrove health assessment.

Thirdly, remote sensing image analysis, synthetic aperture radar interferometry, and machine learning algorithms have proven the effectiveness of extracting mangrove species, leaf area, crown height, and stand biomass in the research of quantitative information extraction of mangrove physiology and biochemistry [36]. The biophysical parameters of mangrove can be extracted in a large area, including height, LAI, stem density, and basal area with the help of ground laboratory data [37,38]. The change in the wetland vegetation community at different times can be obtained by spectral analysis of satellite images of remote sensors with different resolutions [39]. Sentinel series data have reasonable correlation with leaf area index, vegetation coverage, and canopy height. These data can be combined with a machine learning model to predict canopy height [40,41]. Remote sensing data have a significant correlation with canopy height, canopy shape, and height changes [42]. A productivity model based on remote sensing was designed to estimate the light use efficiency (LUE) and primary gross product (GPP) of mangroves in China [43]. Recursive feature is used to select spectral and texture feature variables of vegetation, and random forest and support vector machine algorithm are used as classifiers. The research shows that the combined use of data and methods is helpful for the estimation of mangrove biomass [44]. The relative amounts of morphology, forest age, canopy coverage, aboveground biomass, and wood debris were extracted from the time series data of space-borne optical radar and interferometric radar data [45,46]. The best method to interpret the change in mangrove carbon storage using remote sensing data was found through image processing [47,48]. Using remote sensing technology to extract quantitative information on the physiological and biochemical characteristics of mangrove forests is a practical and effective method. It can reflect indicators such as the photosynthetic activity, leaf area index, and relative productivity of mangrove forests, and thus infer their productivity and growth rate. It can also demonstrate the spatial distribution of water content in forests, revealing their water use efficiency. By combining remote sensing data with ground observation data, such as meteorological station data and soil moisture monitoring data, more accurate physiological and biochemical parameters of mangrove forests can be obtained, such as net photosynthetic rate and water use efficiency. This approach can largely overcome the limitations of traditional quantitative methods in terms of spatial observation range and time scale, enabling a multi-angle understanding of the spatial distribution and dynamic variations of physiological and biochemical information in forests.

Compared with other forest ecosystems, the characteristics of mangrove ecosystems in some aspects are still weak [40]. It will be more difficult to implement effective policies and actions for sustainable protection of mangroves in the context of climate change mitigation and adaptation without effective quantitative methods to monitor the biophysical parameters of mangroves [24]. In this regard, remote sensing is an important tool for monitoring mangroves and determining species and other attributes, and the accurate measurement of species leaf area is crucial for assessing forest growth and health [4,48].

However, there is some uncertainty in the application of quantitative inversion of LAI due to the wide and discontinuous band of multispectral remote sensing data [1]. Hyperspectral data can provide rich and detailed continuous spectral band information [37,46]. With the continuous development of hyperspectral remote sensing technology, a large number of researchers began to retrieve LAI, chlorophyll, and other plant physiological parameters based on hyperspectral remote sensing methods [40,47]. Therefore, the estimation of canopy species abundance based on hyperspectral data and LAI remote sensing retrieval for mangrove communities are of great significance in future forest ecosystem monitoring or research [26,32].

To explore the response mechanism of mangrove leaf area index and main nutrient content in the overall framework, firstly, multi-source data acquisition and pre-processing are carried out, in combination with monitoring indicators such as leaf area index, mangrove canopy leaf water content, chlorophyll a, chlorophyll b, total nitrogen, total phosphorus, total potassium, etc., to analyze the mangrove canopy spectral response characteristics of different species composition and abundance in the study area, and to construct the

extraction method of mangrove canopy species end elements; secondly, the characteristic bands of mangrove evaluation indicators are screened, and the hyperspectral inversion of mangrove monitoring indicators is carried out by using various linear and nonlinear methods, and the best model is selected; finally, the mapping of hyperspectral retrieval results of mangrove canopy indicators was completed. The research results provide basic data and technical support for mangrove ecological remote sensing monitoring [12,28,47].

## 2. Materials and Methods

### 2.1. The Study Area

According to the statistics of 2016, there are approximately 13.6 million hectares of mangroves worldwide, almost one-third of which are located in Southeast Asia, with nearly 20% in Indonesia alone. In the 20 years before 2016, the net loss of mangroves was about 4.3% [42]. Since 2000, more than 60% of the loss in mangroves has been attributed to direct and indirect human activities, mainly occurring in Indonesia, Myanmar, Malaysia, the Philippines, Thailand, and Vietnam [23]. Climate change has also exacerbated natural disasters such as coastal erosion, sea level rise, hurricanes, and droughts, which have caused damage to mangroves. Although mangroves are experiencing continued losses, factors such as sea level rise have led to the expansion of mangrove areas in more and more regions, occupying new sediments or inland areas. Mangroves are the habitat for 341 threatened species and provide a livelihood for over 4.1 million fishermen worldwide. They can avoid property losses of over 65 billion US dollars per year and shelter approximately 15 million people from flooding [15,21].

Haikou Dongzhai Port National Nature Reserve in Hainan Province is selected as the study area (110°32′~110°37′ E, 19°51′~20°1′ N) (Figure 1) [6]. The reserve is the first national nature reserve specifically for mangroves established in 1980 in China. Dongzhai Port Mangrove is the largest coastal beach forest reserve in China, with a total area of 33.38 km$^2$ and a core area of 16.35 km$^2$. The mangrove area is 17.71 km$^2$ and the beach area is 17.59 km$^2$ among them. Dongzhai Port was formed during a major earthquake more than 300 years ago, that is, in 1605. The ditch is full of water and the beach surface is submerged at flood tide; meanwhile, the beach surface is exposed, forming a fragmented swamp beach surface, suitable for the growth of mangroves at ebb tide. The coastal area is brackish marshland, and the water depth of the bay is generally within 4 m. There are 35 species of mangrove plants in 19 families. Among them, *Sonneratia hainanensis*, *Nypa fruticans*, *Sonnerrataovata*, *Sonnerrata paracaseolaris*, *Xylocarpus granatum*, *Rhizophora acemose*, *Acrostichum speciosum wild*, *Sycphilora hydrophylla*, *Barringtonia acemose Thespesia populea*, and *Heritiera littoralis dryand* are rare and endangered mangrove plants in China [17,41].

The distribution patterns of mangrove species in the study area are as follows: *Sonnerata paracaseolaris* grows mostly on sticky coastal soils that are occasionally inundated by tidal waters. *Sonneratia hainanensis* is a tall tree, reaching heights of 5–7 m, while *Nypa fruticans* is a shrub, typically standing 0.5–2.0 m tall. *Sonnerrataovata* communities are found in several areas, mostly within the mid-tide mark, and have well-developed knee-shaped prop roots beneath the forest canopy. *Xylocarpus* granatum grows in low-salinity areas of bays, often on tidal flats that are frequently submerged by seawater. *Rhizophora acemose* scatter along the coast, with gray-green appearances and white mangroves as the dominant species. *Acrostichum speciosum* wild has dense knee-shaped prop roots within the forest, with well-developed branches growing up from the root collar, and grows up to 1.5–4.5 m tall. *Sycphilora hydrophylla* is a shrub community that forms dense plant clumps and is mainly distributed within the high-tide line. They prefer solid, sticky sandy soils with many animal burrows. *Barringtonia acemose* and *Thespesia populea* plants are often flooded by seawater, and their soil type is a fine sandy mud. They are dark green and grow into dense bushes, usually 2.5–3.5 m tall. *Heritiera littoralis dryand* mostly grows on non-floodable coastal beaches, and its soil is composed of fine sandy loam. Due to human activities, most of the mangrove plants in the area have taken on a shrub-like form. It is divided into three sub-regions: A, B, and C in order to compare the indicators of mangroves in

the reserve. Region A is located at the sea outlet, close to the urban area and aquaculture area, with intensive human activities; Region B has the least human development activities and is basically unaffected by urban activities; Region C is surrounded by agricultural production areas with intensive farmland. There is reason to believe that although the mangrove species are close to the seawater environment, there are certain differences in the key indicators of mangrove in the three regions [21].

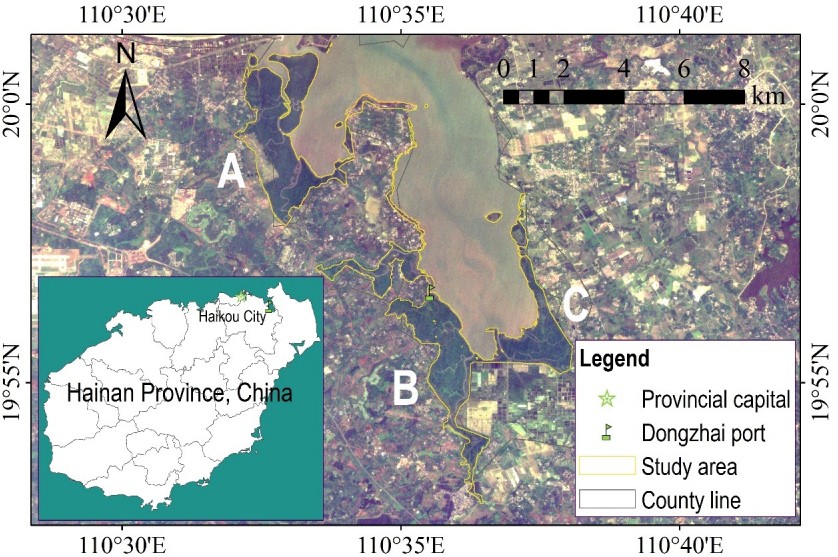

**Figure 1.** Location of mangrove growing area in Dongzhai port. The study area is located 30 km east of Haikou, the capital of Hainan Province. The well-growing mangroves are 3–4 m high. Mangroves are rooted on the beach by the crisscross and developed pillar roots, respiratory roots, and aerial roots at the base of the trunk.

### 2.2. Data Sources and Processing

The hyperspectral data of OHS satellite that passed the zenith on 11 November 2021 were obtained. OHS adopts push-scan imaging mode, with a single imaging range of 150 km × 3200 km, spatial resolution of 10 m, and spectral resolution of 2.5 nm. The wavelength range is 400~1000 nm. The number of bands is designed to transmit 32 due to the limitation of compression and storage. A single hyperspectral satellite can orbit the Earth about 15.16 times a day, and the single data acquisition time of each orbit does not exceed 8 min. At present, the revisit period of a single satellite is 6 days, and that of four satellites is 2 days [22].

The data are processed according to the process of radiometric calibration [16], atmospheric correction [40], orthophoto correction [44], and study area clipping in order to obtain the true reflectance of the mangrove canopy. The metadata are read and processed to obtain the center wavelength, half-height wave width, calibration coefficient, and coordinate information of each spectral band. The gray value is converted into absolute radiance, and the digital quantization value is converted into the atmospheric apparent radiance value with physical significance to determine the accurate radiance value at the pupil of the sensor [2,29]. The FLAASH algorithm is used to correct the data in the atmosphere in order to eliminate the radiation error caused by atmospheric absorption and obtain the true reflectivity [29]. The growth height of mangroves in the study area is quite different. With the help of digital surface model data, the obvious image distortion caused by plant height and camera geometric characteristics will be processed [23]. The study used ASTER GDEM 30 M resolution digital elevation data (from the Chinese Geospatial Data Cloud website: https://www.gscloud.cn/, accessed on 11 November 2021). This cut out the extraterritorial data, and formed a hyperspectral image of the true reflectance of the mangrove.

Seven parameters of typical mangrove were synchronously measured, namely leaf area index, water, chlorophyll A, chlorophyll B, total nitrogen, total phosphorus, and total potassium on the day of satellite data acquisition (Table 1). A total of 20 points of laboratory data were obtained [41]. The PSI Laipen LP110 leaf area index measuring instrument produced in the Czech Republic is adopted for leaf area index [1]. The instrument uses a wide-angle optical sensor to measure solar radiation and then calculate LAI and other canopy structure indicators [37]. The light interception of the canopy is determined by five different angles of measurement above and below the canopy, and the data are substituted into the vegetation canopy radiation transfer model to calculate LAI (Figure 2) [40]. After the measurement, the leaves are sampled and sealed in a polyethylene bag. After recording the metadata information, the leaves are sent to the Chemical Laboratory of Hainan University for nutrient analysis. The moisture is measured with a VM-E10 halogen moisture tester, and the average value is taken after 10 measurements; chlorophyll A and chlorophyll B are measured by a spectrophotometer. The determination principle is that they have the maximum absorption values at 645 nm and 663 nm. The content value can be calculated according to the empirical formula; total nitrogen was determined by UNICUBE trace element analysis; total phosphorus and total potassium were determined by Axios-type X-ray fluorescence spectrometry [46].

**Table 1.** Statistics of the contents of seven mangrove parameters in each leaf sample obtained after in situ detection. The dimensions of each leaf sample were 0.2 m × 0.2 m. The weight is 0.2 kg, which is enough for mangrove parameter analysis and verification.

| Serial Number | Value | Mangrove Parameters | | | | | | |
|:---:|:---:|:---:|:---:|:---:|:---:|:---:|:---:|:---:|
| | | LAI | MSTR | Chla | Chlb | TN | TP | TK |
| 1 | Minimum | 1.46 | 38.70 | 308.01 | 120.24 | 23.55 | 1.67 | 0.73 |
| 2 | Maximum | 6.24 | 77.00 | 1004.46 | 348.79 | 84.69 | 7.44 | 6.63 |
| 3 | Average | 4.01 | 62.79 | 696.17 | 182.26 | 45.56 | 3.72 | 3.17 |
| 4 | Standard deviation | 1.36 | 7.92 | 175.21 | 52.37 | 21.01 | 1.82 | 1.81 |

Note: The unit of LAI is $m^2 \ m^{-2}$; the unit of MSTR is %; the unit of Chla, Chlb, TN, TP, and TK is g $kg^{-1}$.

The hyperspectral data of these 20 sampling points were synchronously measured in the field using the PSR-1100 portable ground object spectrometer [14]. The wavelength range of the instrument is 325–1075 nm, and the spectral resolution is 3 nm. The reflectivity data with a wavelength of 400–1000 nm are selected in order to match the OHS satellite data (Figure 3) [25]. Ground spectral data, which are very close to the leaves and suffer less atmospheric interference, can generally be used to correct satellite data. Synchronously, these 20 sets of data are also required for modeling, and play a good role in discovering characteristic bands.

*2.3. Research Method*

In order to highlight the effective information in the spectral data, especially the weak information, the original spectral data are transformed into 23 types to obtain a total of 24 groups of spectral input data (Table 2). By converting the original reflectivity, a series of reflectivity independent variables can be formed, which can amplify or reduce the reflectivity value of the characteristic peak and improve the probability of spectral recognition [24]. When establishing the regression model with the physical and chemical composition analysis data, the matching relationship between the spectral data and the laboratory data is analyzed through the comprehensive validation of various methods [33].

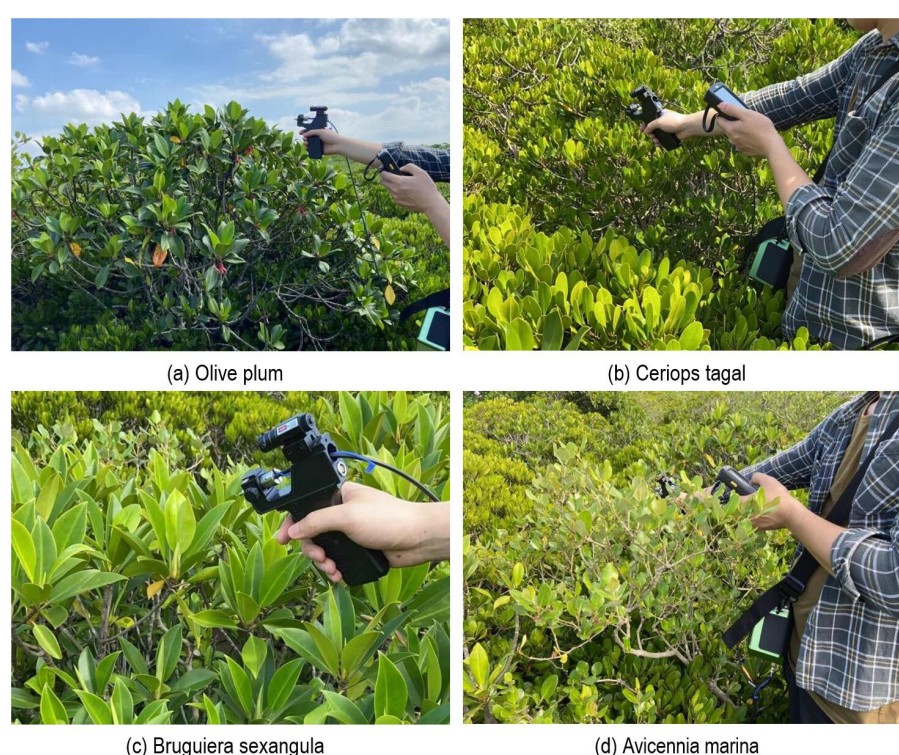

**Figure 2.** Typical spectral data acquisition process. Each branch has three consecutive scans of different angles, and the mean value is taken as the final reflectance spectrum of the branch leaves. The spectrum can be directly collected, and the optical fiber probe can be used to collect the spectral information at different canopy azimuth angles when the height of the plant is lower than the height of the operator. (**a**) Olive plum; (**b**) Ceriops tagal; (**c**) Bruguiera sexangula; (**d**) Avicennia marina.

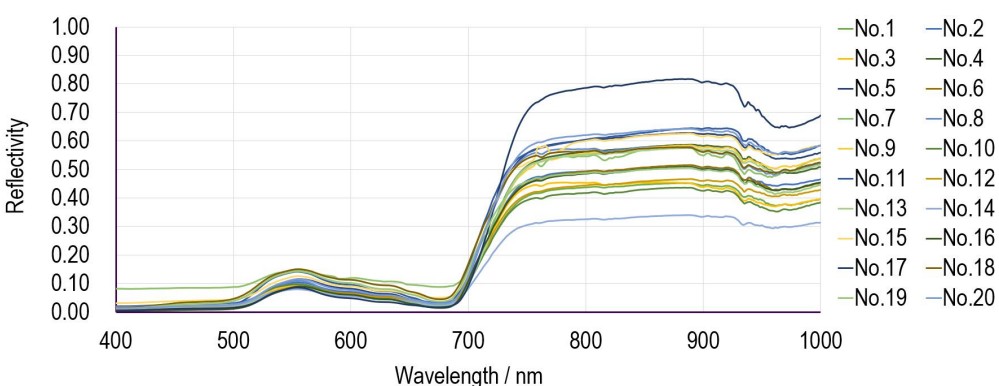

**Figure 3.** Canopy reflectance data of 20 mangrove sampling points collected by hand-held spectrometer.

**Table 2.** Spectral transformation methods and calculation formulas. The collected spectral data are processed by a series of spectral transformations to highlight the effective information expression in the spectral data.

| Serial Number | Transformation Method | Process Formulas |
|---|---|---|
| 1 | Original spectrum | $X_i = R_i$ |
| 2 | Exponential | $X_i = e^{R_i}$ |
| 3 | Multiple scattering correction | $X_i = (R_i - b_i)/k_i$ |
| 4 | Envelope elimination | $X_i = R_i/C_i$ |

**Table 2.** *Cont.*

| Serial Number | Transformation Method | Process Formulas |
|---|---|---|
| 5 | Logarithm | $X_i = \mathrm{Ln}\,(R_i)$ |
| 6 | Homogenization | $X_i = (R_i - R_{min})/(R_{max} - R_{min})$ |
| 7 | First-order differential | $X_i = R_i{}'$ |
| 8 | Second-order differential | $X_i = R_i{}''$ |
| 9 | Third-order differential | $X_i = R_i{}'''$ |
| 10 | Exponential after first-order differential | $X_i = e^{R_i{}'}$ |
| 11 | Exponential after second-order differential | $X_i = e^{R_i{}''}$ |
| 12 | Exponential after third-order differential | $X_i = e^{R_i{}'''}$ |
| 13 | Logarithm after first-order differential | $X_i = \mathrm{Ln}\,(R_i{}')$ |
| 14 | Logarithm after second-order differential | $X_i = \mathrm{Ln}\,(R_i{}'')$ |
| 15 | Logarithm after third-order differential | $X_i = \mathrm{Ln}\,(R_i{}''')$ |
| 16 | Homogenization after first-order differential | $X_i = (R_i{}' - R'_{min})/(R'_{max} - R'_{min})$ |
| 17 | Homogenization after second-order differential | $X_i = (R_i{}'' - R''_{min})/(R''_{max} - R''_{min})$ |
| 18 | Homogenization after third-order differential | $X_i = (R_i{}''' - R'''_{min})/(R'''_{max} - R'''_{min})$ |
| 19 | Envelope elimination after first-order differential | $X_i = R_i{}'/C_i$ |
| 20 | Envelope elimination after second-order differential | $X_i = R_i{}''/C_i$ |
| 21 | Envelope elimination after third-order differential | $X_i = R_i{}'''/C_i$ |
| 22 | Multiple scattering correction after first-order differential | $X_i = (R_i{}' - b_i)/k_i$ |
| 23 | Multiple scattering correction after second-order differential | $X_i = (R_i{}'' - b_i)/k_i$ |
| 24 | Multiple scattering correction after third-order differential | $X_i = (R_i{}''' - b_i)/k_i$ |

Note: $X_i$ is the processed spectral reflectivity; $R_i$ is the spectral reflectivity; $i$ is the band variable; $b_i$ is the baseline offset; $k_i$ is the baseline translation; $R_{min}$ is the minimum reflectivity; $R_{max}$ is the maximum reflectivity; and $C_i$ is the envelope curve value.

Spectral characteristic values are obtained on the basis of spectral transformation and spectral feature extraction, and the mathematical model of content and spectral characteristic values is established. Three band combination algorithms, band difference, band ratio, and band difference and ratio are selected for content inversion and accuracy evaluation in order to facilitate the understanding and understandability of the model [42]. The band difference model can remove the interference information and obtain the spectral characteristic variable most closely related to the content. The calculation formula is

$$y = a(F_1 - F_2) + b, \tag{1}$$

where $y$ is the inversion value of mangrove content; $a$ and $b$ are model coefficients, respectively; $F_1$ is the characteristic variable most relevant to the content; $F_2$ is the characteristic variable most irrelevant to the content. The band ratio model plays a more significant role in amplifying favorable features and suppressing unfavorable features [22]. It can also further remove the multiplicative error between features and improve the inversion accuracy of the model on the basis of enhancing spectral features. The formula is

$$y = a\left(\frac{F_1}{F_2}\right) + b, \tag{2}$$

where $y$ is the inversion value of mangrove content; $a$ and $b$ are model coefficients, respectively; $F_1$ and $F_2$ is an arbitrary combination of two spectral characteristic variables. The band difference and ratio are the most classical spectral content calculation models,

which can amplify spectral characteristics and remove certain system errors, and have the advantages of the above two models. The calculation formula is

$$y = a\left(\frac{F_1 - F_2}{F_1 + F_2}\right) + b, \tag{3}$$

where $y$ is the inversion value of mangrove content; $a$ and $b$ are model coefficients, respectively; $F_1$ and $F_2$ is an arbitrary combination of two spectral characteristic variables.

*2.4. Accuracy Evaluation*

The coefficient of determination ($R^2$) of precision evaluation is to evaluate the overall prediction ability of the model. If the coefficient of determination $R^2$ calculated by the model is closer to 1, the precision of the model is higher, and the explanatory power of the variables in the model to the dependent variables is also higher [49]. The calculation formula is as follows:

$$R^2 = 1 - \frac{\sum_{i=1}^{n}(y_i - \hat{y}_i)^2}{\sum_{i=1}^{n}(y_i - \overline{y})^2}, \tag{4}$$

where $n$ is the sample size, $y_i$ is the assay value of the content of point $i$, $\hat{y}_i$ is the content prediction value of spectral method of point $i$, and $\overline{y}$ is the mean of the assay value of the samples.

The mean relative error (*MRE*) evaluates the accuracy of the prediction results of the model, and calculates the mean deviation of the prediction results from the true value, reflecting the accuracy of the prediction results of the model [50]. The lower the value is, the closer the prediction value of the model is to the true value. The calculation formula is as follows:

$$MRE = \frac{1}{n}\sum_{i=1}^{n}\frac{|y_i - \hat{y}_i|}{\hat{y}_i}, \tag{5}$$

where $n$ is the sample size, $y_i$ is the assay value of the content of point $i$, and $\hat{y}_i$ is the content prediction value of the spectral method of point $i$.

Root means square error (*RMSE*) represents the degree of dispersion of the model prediction results compared with the true value of the dependent variable, reflecting the stability of the model prediction performance [51]. The lower the value, the better the stability of the model prediction results. The calculation formula is as follows:

$$RMSE = \sqrt{\frac{1}{n}\sum_{i=1}^{n}(y_i - \hat{y}_i)^2}, \tag{6}$$

where $n$ is the sample size, $y_i$ is the assay value of the content of point $i$, and $\hat{y}_i$ is the content prediction value of the spectral method of point $i$.

Generally, the closer the slope of $R^2$ and fitting equation is to 1, the smaller the root mean square error (*RMSE*) and mean relative error (*MRE*) are, the higher the accuracy of the model is, and the more similar the trend of the prediction result is to the real situation. This study screened the best model in different models based on $R^2$, and evaluated the accuracy and stability of the model through *MRE* and *RMSE* in order to avoid the overfitting phenomenon of the training model to misjudge the results [52].

## 3. Results

*3.1. Calculation Results and Mapping*

The above algorithm is implemented, and the precision evaluation results are obtained in Python (Table 3). Inversion models of LAI and Chla are similar in structure. Both need to calculate the index after high-order differential of the spectrum. The $R^2$ coefficient reaches above 0.65, and the model is effective. The envelope needs to be calculated after the second-order differential of the spectrum for the inversion of MSTR and Chlb. The established models are linear models. This indicates that leaf moisture is mainly related to

Chlb, and R$^2$ coefficient has exceeded 0.80, which is significantly higher than the calculation accuracy of other components [40]. The modeling spectra of TN and TP are the simplest. The optimal model can be obtained by using the original spectra or exponential changes directly. However, the accuracy of the model does not exceed 0.60 due to the low content. The spectral transformation method adopted by TK is homogenization after first-order differential, and the model is a linear regression equation. The accuracy exceeds 0.70, and MRE and RMSE are also very low [48].

**Table 3.** Calculation model and precision evaluation results of mangrove key indicators.

| Serial Number | Indicators | Transformation Method | Calculation Model | R$^2$ | MRE | RMSE |
|:---:|:---:|:---:|:---:|:---:|:---:|:---:|
| 1 | LAI | Exponential after third-order differential | $y = -0.39\frac{b_{910}-b_{772}}{b_{910}+b_{772}} + 4.61$ | 0.66 | 0.18 | 0.83 |
| 2 | MSTR | Envelope elimination after second-order differential | $y = 622.31b_{943} + 480.94b_{975} + 60.39$ | 0.92 | 0.01 | 0.53 |
| 3 | Chla | Exponential after second-order differential | $y = -267\frac{b_{552}-b_{917}}{b_{552}+b_{917}} + 964.20$ | 0.75 | 0.12 | 113.26 |
| 4 | Chlb | Envelope elimination after second-order differential | $y = -117.46b_{565} + 112.43b_{681} + 185.29$ | 0.83 | 0.04 | 9.64 |
| 5 | TN | Original spectrum | $y = -14.23(b_{744} - b_{904}) + 57.17$ | 0.56 | 0.29 | 12.97 |
| 6 | TP | Exponential | $y = -3.71\frac{b_{722}-b_{934}}{b_{722}+b_{934}} + 4.89$ | 0.57 | 0.27 | 1.19 |
| 7 | TK | Homogenization after first-order differential | $y = 56.09b_{864} + 67.76b_{939} + 5.22$ | 0.73 | 0.027 | 0.084 |

Note: $b_i$ is the processed or original spectral data, and $i$ is the corresponding wavelength (nm).

The distribution of the content of seven mangrove components presents different laws according to the analysis of the calculation results (Figure 4) [16]. The distribution of LAI shows a trend of higher content as it is closer to the water surface. This phenomenon presents similar laws in the river and sea [21]. In particular, the LAI content in region A, which is similar to the peninsula shape, is significantly higher than that in the other two regions. The distribution of MSTR is contrary to that of LAI, and the closer to the land, the higher the moisture of leaves [47]. This is related to the transpiration of mangroves. The data acquisition time is at noon. The photosynthesis of leaves means that the larger the leaf area, the lower the moisture content. The contents of Chla and Chlb present basically the same distribution pattern. The content has reached a high level in the distribution area of mangroves, which reflects the good growth state of mangroves in the reserve. The distribution of TN is significantly different from that of TP and TK. The content is high along the intersection area of mangrove and water [4]. The content distribution of TP is low and TK is similar. The denser the vegetation is, the more enriched the element content is.

### 3.2. The Strength of Nutrient Content along the Three Regions

The result of mean percentage shows that there is no significant difference in the retrieval index of mangroves in the three regions (Figure 5a), which shows that although the environment of the three regions is different, the overall health level of mangroves is the same [34,53]. The maximum value calculation results indicate that the maximum values of the other six indicators appear in region A and B, except for the maximum value of Chlb in region C. The index with a large difference is TK, and the maximum values of other indicators are also approximately the same (Figure 5b) [21]. STD indicates the mutation in the content of components in different regions, and MSTR and TN are the two mangrove indicators with the most drastic changes. The maximum STD of the other six indicators appear in the other two regions, except for the maximum STD of TN that appears in region B (Figure 5c) [4]. The total content of various components in region A is significantly higher than the other two regions, except for the total potassium content (Figure 5d).

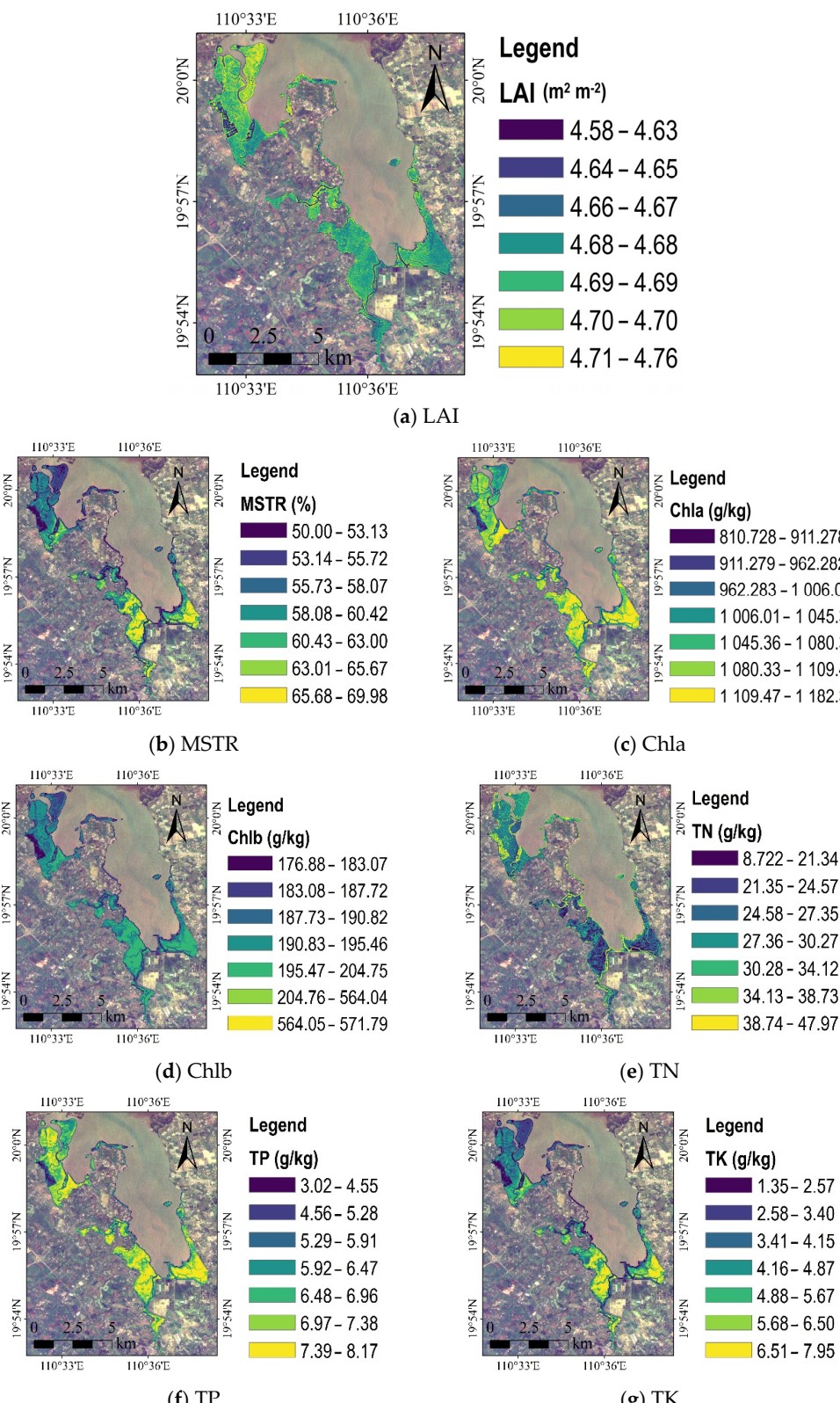

**Figure 4.** Mapping results of 7 components. (**a**) The distribution of LAI is generally lower than 4.76 m$^2$ m$^{-2}$; (**b**) the distribution of MSTR is lower than 70.00 g kg$^{-1}$ as a whole; (**c**) the average value of Chla is 1061.19 g kg$^{-1}$; (**d**) the average value of Chlb is 191.99 g kg$^{-1}$, and its distribution is significantly related to Chla; (**e**) the content of TN is lower than 48.00 g kg$^{-1}$ in most areas; (**f**) the average content of TP is 6.76 g kg$^{-1}$; (**g**) the average content of TK is 4.40 g kg$^{-1}$, and the distribution has a certain correlation with the TP.

### 3.3. Distribution Patterns of Nutrient Content in Mangrove along the LAI

Covariance can reflect the change trend among multiple inversion results of mangrove (Table 4) [33]. If the distribution law of a nutrient element and LAI is consistent, the difference of the covariance is positive; if the distribution law is opposite, the covariance difference is negative [49]. When the covariance is zero, there is no significant decisive relationship between the two. The results showed that the covariance between LAI and all six leaf nutrient components was small, with a negative correlation with TN and positive correlation with Chla. Chla and Chlb were the most significant components with positive correlation of covariance, and the covariance reached 45.05; the second most significant were MSTR and Chla, with a covariance of 34.23. The most significant negative correlation of covariance was Chla and TN, with a covariance of −35.97. Chla is the key identification component of mangrove growth and health.

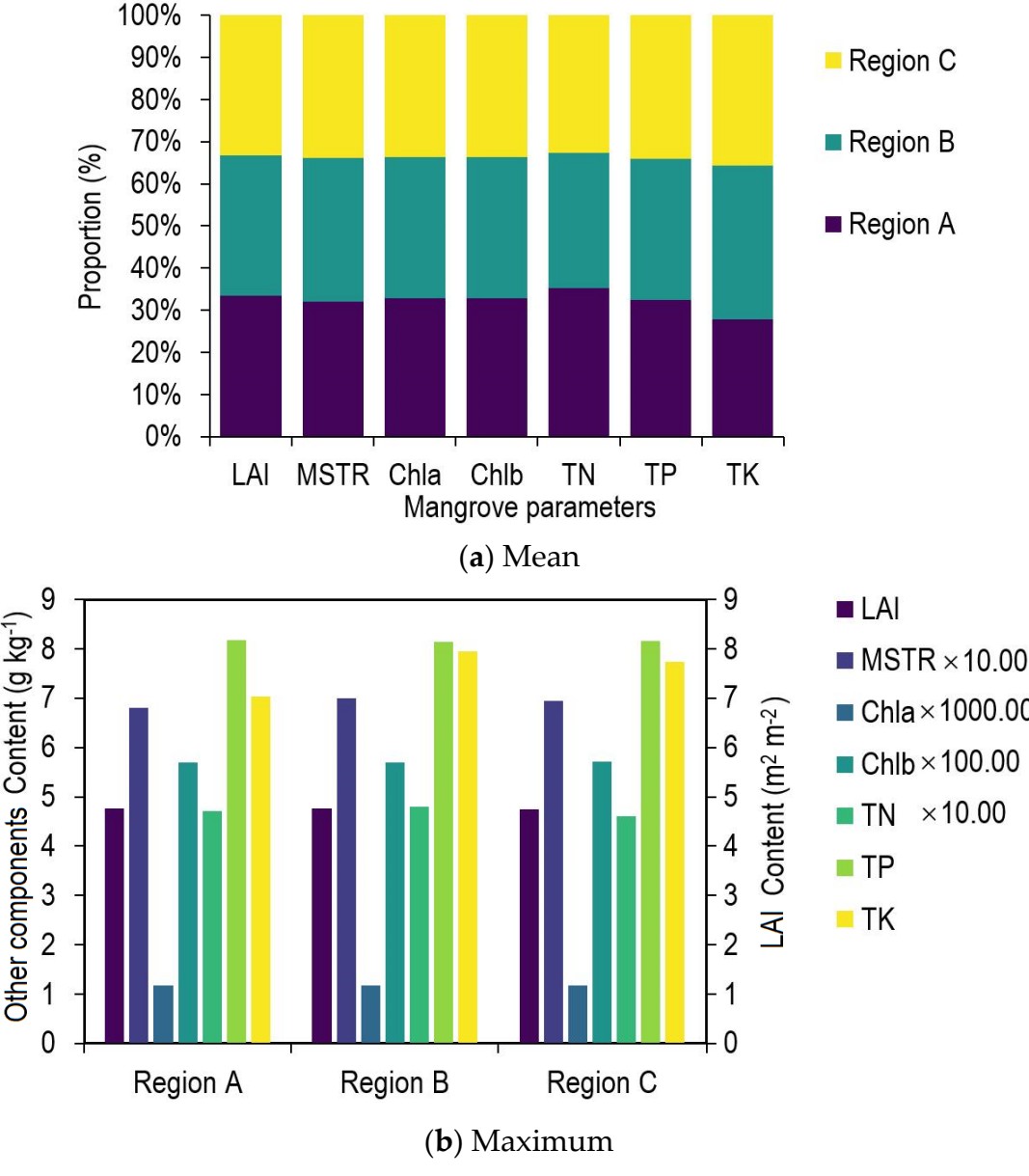

(**a**) Mean

(**b**) Maximum

**Figure 5.** *Cont.*

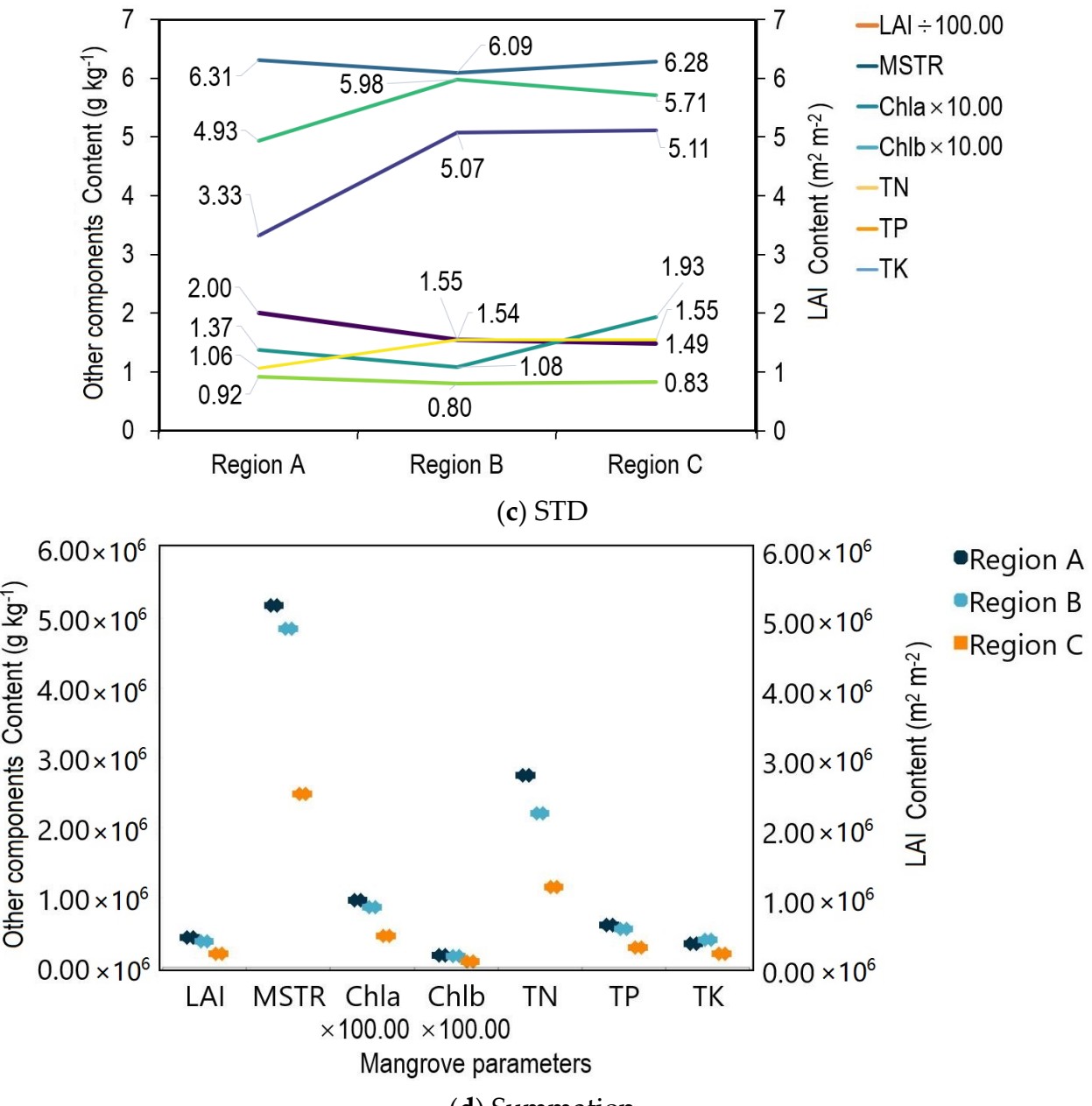

**Figure 5.** Statistical results of mangrove main nutrient contents in different regions. (**a**) There is no significant difference in the average nutrient content of the three regions, which is in the same order of magnitude; (**b**) the maximum values of the other six substances in the three regions is close in addition to the content of total potassium; (**c**) the maximum STD values of LAI and total nitrogen are large, and the other five substances have little difference in the three regions; (**d**) in terms of total content, the content of various components in region A is significantly higher than that in the other two regions. However, the content of total potassium in region B is the highest.

The correlation coefficient can be used to further analyze the relationship between the retrieval results due to the different order of mangrove nutrient composition [54]. The correlation coefficient is a special covariance after standardization, which is calculated by dividing the covariance by the standard deviation. Then, the correlation coefficient between mangrove nutrient elements and LAI is calculated (Table 5) [55]. The covariance range is from positive infinity to negative infinity, and the correlation coefficient can only change between $\pm 1$ [49]. It is concluded that there is a significant negative correlation between LAI and TN, with a coefficient of $-0.67$. The correlation coefficient with the content of

the other five nutrient elements is positive, but lower than 0.40. The contents of nutrient elements with a correlation coefficient exceeding 0.80 include MSTR and TK (0.98), Chla and TP (0.96), Chla and TK (0.87), MSTR and Chla (0.86), MSTR and TK (0.83), and MSTR and TP (0.81). It is worth noting that TN has a negative correlation with all components, and the negative correlation with TP, MSTR, and Chla has exceeded 0.75. Chlb is a very special nutritional component, and its content has no significant correlation with LAI and the other five nutritional components.

**Table 4.** Covariance matrix of nutrient contents and LAI. The covariance can be positive or negative. A positive value indicates that two variables have a positive correlation, i.e., when one variable increases, the other variable also increases; a negative value indicates that two variables have a negative correlation, i.e., when one variable increases, the other variable decreases.

|      | LAI   | MSTR  | Chla   | Chlb  | TN     | TP    | TK    |
|------|-------|-------|--------|-------|--------|-------|-------|
| LAI  | /     | 0.00  | 0.05   | 0.00  | −0.01  | 0.00  | 0.00  |
| MSTR | 0.00  | /     | 34.23  | 3.55  | −2.76  | 0.45  | 0.93  |
| Chla | 0.05  | 34.23 | /      | 45.05 | −35.97 | 7.01  | 10.81 |
| Chlb | 0.00  | 3.55  | 45.05  | /     | −3.31  | 0.60  | 1.10  |
| TN   | −0.01 | −2.76 | −35.97 | −3.31 | /      | −0.47 | −0.87 |
| TP   | 0.00  | 0.45  | 7.01   | 0.60  | −0.47  | /     | 0.14  |
| TK   | 0.00  | 0.93  | 10.81  | 1.10  | −0.87  | 0.14  | /     |

**Table 5.** Correlation matrix between nutrient contents and LAI. The correlation coefficient is a statistical measure used to assess the linear relationship between two variables and reflects whether the two variables change in the same direction. The correlation coefficient ranges from −1 to 1, with larger absolute values indicating a stronger relationship between the two variables.

|      | LAI   | MSTR  | Chla  | Chlb  | TN    | TP    | TK    |
|------|-------|-------|-------|-------|-------|-------|-------|
| LAI  | /     | 0.20  | 0.34  | 0.10  | −0.67 | 0.34  | 0.22  |
| MSTR | 0.20  | /     | 0.86  | 0.40  | −0.78 | 0.81  | 0.98  |
| Chla | 0.34  | 0.86  | /     | 0.38  | −0.76 | 0.96  | 0.87  |
| Chlb | 0.10  | 0.40  | 0.38  | /     | −0.31 | 0.37  | 0.40  |
| TN   | −0.67 | −0.78 | −0.76 | −0.31 | /     | −0.72 | −0.79 |
| TP   | 0.34  | 0.81  | 0.96  | 0.37  | −0.72 | /     | 0.83  |
| TK   | 0.22  | 0.98  | 0.87  | 0.40  | −0.79 | 0.83  | /     |

## 4. Discussion

Mangrove is a special vegetation type that grows in the upper part of the intertidal zone of tropical and subtropical coasts, and is an ecological key area in the transition between land and sea, with unique hydrological characteristics, biogeochemical, and ecological functions [4,9,56]. Hainan Province is one of the areas with the widest distribution of mangroves and the richest biodiversity in China [17,41]. LAI represents the density of vegetation leaves, and is the key factor affecting the photosynthetic effective radiation capacity of the canopy in the carbon cycle. Its level directly affects the strength of the photosynthetic capacity, and has an important impact on the global carbon cycle and vegetation growth and development [37,46,47]. Accurately grasping the mangrove LAI represents the basic work undertaken to evaluate the vegetation growth status in the land and sea transition zone. The relevant variables of mangrove LAI can be used as the health indicators of the forest ecosystem. The ability of LAI to characterize the canopy structure is crucial to understand the LAI in assessing the health status, predicting future growth, and mangrove production.

Satellite, airborne, and ground remote sensing sensors are used to receive the reflected signals of ground objects. Different mangrove tree species have different absorption and reflection characteristics of electromagnetic waves of different wavelengths, forming the

characteristic spectrum of mangrove reflectivity changing with wavelength [19]. The mangrove spectrum has a fingerprint effect on mangrove classification and target recognition, which is a bridge connecting remote sensing theory and remote sensing application. Spectral data sets of spectra and characteristic parameters that can cover a variety of typical targets are formed by collecting, processing, and analyzing the measured spectra of typical mangroves [20].

Mangroves are affected by the special growth environment, and the traditional survey technology is faced with many challenges [10]. Hyperspectral technology can obtain the nutrient content of mangrove plants from both mechanism and statistics. Satellite data have wide coverage, strong timeliness, and high spectral resolution, but limited spatial resolution, so are very suitable for large-scale regional surveys, covering regional, national, and even global scales [35]. The spatial and spectral resolution of airborne hyperspectral are high, but the data acquisition conditions and costs are high, so it is suitable for a large-scale survey in key areas [10]. The UAV hyperspectral data acquisition method is very flexible, but the data acquisition efficiency is low, so it is suitable for a small-scale survey or field test [50]. The ground hyperspectral has the highest spectral resolution, but there is no image information, and the data are in the form of scattered points, which are suitable for data modeling, ground experiments, and verification [44]. Through the cooperation of satellite hyperspectral remote sensing and ground hyperspectral data, this research obtains the data of photoelectric detection data to achieve more accurate quantitative remote sensing [38,45,53].

This paper focuses on two methods in order to realize the effective assimilation of satellite data and ground data [46,47]. The first one is to determine the quantitative relationship between mangrove monitoring indicators and characteristic bands to achieve the extraction of nutrient content. Through processing the spectral data, including itself and its 23 kinds of transformation data, three band combination algorithms of band difference, band ratio, and band difference and ratio are tested to form a controllable machine learning model package [12,26,39]. They are used to conduct large-scale training and learning on mangrove component content and hyperspectral data, and extract information from statistical significance. Although this method cannot explain the basis of feature band selection from the mechanism, it is effective in a certain region and a certain period of time [11,44].

The research reveals that, firstly, different spectral transformation and band combination models are needed for the extraction of different components of mangrove [10,20,22,47]. The processed data including spectral differentiation and de-enveloping can significantly improve the regression accuracy of the model. This modeling method is very accurate since spectral data are measured on the ground and in situ data are obtained; secondly, the numerical comparison of the nature reserve in different regions shows that the growth status of mangroves in the three regions is consistent, which shows the local importance to the nature reserve [23,24]. According to the satellite monitoring results, it is not found that human activities have affected the growth of mangroves in different regions. Thirdly, the study confirmed that LAI, as a geometric index of leaves, has no decisive effect on the nutrient composition of mangroves. Through covariance and correlation analysis, although LAI has little relationship with the content of nutrients, there is a significant correlation between the six nutrients [37,40,46,47]. This conclusion can not only guide the scientific evaluation of mangrove growth quality, but can also control the potential risk sources to guide the official work.

Recently, hyperspectral sensors based on satellites, aircraft, unmanned aerial vehicles, and the ground have emerged endlessly, and have been increasingly introduced into mangrove ecological assessment [12,21,42,44,47]. The traditional sampling and analysis technology has been unable to meet the needs of digital applications due to the special growth environment of mangroves, and the intervention of new technologies is urgently needed [14]. In the future, the research focus of hyperspectral technology in this field should be to form the mangrove basic spectral database, realize the scientific modeling of

hyperspectral data, and solve the current regional and temporal constraints. Spectral technology is one of the important development directions in this field with the accumulation of data. As this region has been designated as a nature reserve by China, all the mangroves here are in very good condition. Among the mangrove forests in China, the mangroves of the study area are in one of the largest contiguous areas, with the most diverse tree species, best forest quality, and richest biodiversity, and the area has been listed as an internationally important wetland site. This indicates that the overall health level of the mangroves in Dongzhai Port is relatively high. Our field investigations have also confirmed this, with all the mangroves being classified as being in normal condition.

## 5. Conclusions

Mangroves are an ecosystem with rich biodiversity. They are not only an important habitat for rare and endangered waterfowl but also a habitat and breeding ground for many marine and coastal organisms [16,31]. In China, the number of mangrove species is also abundant, with 37 species belonging to 20 families and 25 genera. There are many different kinds of animals in the mangrove ecosystem, including fish, shrimp, crabs, clams, and birds. These animals depend on the environment in which mangroves grow and form a complex food chain, maintaining the balance of the ecosystem. In addition, mangroves have protective functions, such as purifying seawater, preventing wind and waves, and carbon fixation [48]. It should be noted that due to human interference and destruction, the biodiversity of some mangrove areas is threatened, making the protection and restoration of mangroves particularly important [10,46,48]. As far as we know, the current study quantifies the relationship between different LAI and leaf nutrient content of mangrove for the first time [1]. In general, our research can improve our understanding of the relationship between key components in the mangrove growth process from the perspectives of moisture content, leaf biology, chemical elements, etc. The research shows that although LAI is not enough to dominate the spatial distribution of mangrove nutrient content, there is a significant correlation between different nutrient components. The relationship between all six nutrients was analyzed, and the grade relationship of mangrove elements in the study area was accurately evaluated in order to evaluate the degree of the close relationship. The above conclusions are drawn due to the data of only one period. The factors such as sea water tide level and mangrove biology have not been fully considered. Further research is needed to assess whether LAI and the six nutrients change over time, and how they affect the ecosystem function of mangrove.

**Author Contributions:** X.C. and Y.Y.: methodology, software, and writing—original draft preparation; D.Z.: conceptualization, writing—review and editing, project administration, and funding acquisition; X.L. and Y.G.: investigation, data curation, and visualization; L.Z. and D.W.: writing—review and editing, supervision, and project administration; J.W. (Jianhua Wang), J.W. (Jin Wang) and J.H.: data curation, writing—review and editing, visualization, and software. All authors have read and agreed to the published version of the manuscript.

**Funding:** This research was funded by the special technical innovation project of provincial scientific research institutes (jscx202023), the National Natural Science Foundation of China (Grant No. 42272346), and the National Key Research and Development Program of China (No. 2022YFB3902000, No. 2022YFB3902001).

**Data Availability Statement:** The data and algorithm code presented in this study are available on request from the corresponding author.

**Conflicts of Interest:** The authors declare no conflict of interest.

## Abbreviations

| | |
|---|---|
| OHS | Orbita HyperSpectral |
| FLAASH | Fast line-of-sight atmospheric analysis of spectral hypercubes |
| PSI | Photon Systems Instruments |
| LAI | Leaf area index |
| MSTR | Moisture |
| Chla | Chlorophyll a |
| Chlb | Chlorophyll b |
| TN | Total nitrogen |
| TP | Total phosphorus |
| TK | Total potassium |
| STD | Standard Deviation |
| UAV | Unmanned aerial vehicle |

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
