# Peer review of "Response Mechanism of Leaf Area Index and Main Nutrient Content in Mangrove Supported by Hyperspectral Data"

_forests, doi:10.3390/f14040754_

Round 1
Reviewer 1 Report
1 The importance of studying mangrove forests in general does not follow from the introduction. Information on their density in the world and in China should be added. It is also necessary to describe their uniqueness in terms of production process, biodiversity.
2. In general, in the introduction, we would like to see concrete facts and knowledge, not only references to sources.
3 The authors talk a lot about the classification of mangrove forests based on satellite images. But the reader wants to know right away the specific signs of mangrove forest decoding - are these water level fluctuations? Phenological features? any signs of texture?
4. The article says nothing (or very little) about the biology and ecology of these communities: species composition, cover parameters that distinguish mangroves from other forests, etc. It is fashionable to make it brief.
5. Areas are probably better given in hectares rather than km2.
Which digital elevation model was used in the analysis? (resolution)
6. The authors write that LAI has no units of measurement. In fact, it follows from the definition of the index that it is measured in m2/m2 or ha/ha.
7. Forest characteristics should be added to the Study Area section, if such parameters are known. Height, number of species, trunk diameter, age, biomass.
8. What is the significance of Fig. 5А? Relative values by site (ABC) are summarized in one column. It does not make sense. It is better to separate them.
9. The units shown in the figures are g/kg. LAI has other units. For LAI you have to make a second scale to the right.
10. The lines in Figure5B are better turned into graphs. Because there is no gradient between ABC.
11. There are a lot of blue lines here - better to use contrasting colors.
12. Is it possible to perform a field verification of the method in the study? That is, to check in natural conditions the values of the coefficients predicted by the model.
13. Figures 6 and 7 are tables, not figures.
14. The results state that the parameters obtained allow us to estimate the physiological state of the forests. It would be good to create a gradation of parameters and divide forests into categories - normal, weakened, degraded, etc.
Reviewer 2 Report
The manuscript presents Response Mechanism of Leaf Area Index and Main Nutrient Content in Mangrove Supported by Hyperspectral Data, which is interesting. It is relevant and within the scope of the journal.
I gave some suggestion to complete my review comments below:
Abstract:
The abstract section needs to complete with more information. The abstract should be improved.
Introduction:
The literature review is weak some information will add then it will be better.
Materials and Methods:
Study area:
Would you please explain more the study area?
Result and Discussion:
The rest of the paper is very well written and some more points can be added in the conclusion part.
The paper may be accepted after modification as suggested.

Reviewer 3 Report
The paper applied Leaf Area Index (LAI) and measurements of selected nutrient to assess the health of mangroves in Hainan Province, China. The importance of the paper is based on its link between the LAI and the key components of the mangrove growth. The paper provides important information with potential applications in the field of environmental monitoring and ecological assessment.
However, the paper could benefit from the following comments/suggestions:
Abstract:
The abstract is typically standalone part of the paper. Even thought there is a part of the abbreviations at the end of the paper, there might be a need to include the terminology in full in the abstract. Please check of the terminology in the abstract can be understood without referring to the paper. Please check the use of (is vs was) in the abstract and throughout the paper. Typically, when referring to the method (was) is used. Please delete (etc.). Please revisit (for the first time) in both in the abstract and conclusion.
Introduction:
The introduction is sound and provide coherent background relevant to the study. I will recommend revisiting the some sentences (gerund; without verb). Please check the (And) at the beginning of the sentences.
The last part of the introduction provides a justification of the study leading the aim. However, there is a need for the objectives of the study to be more pronounced (starting in the paragraph ‘In the overall framework..’ page 3.
Material and methods:
The procedures is scientifically sound, and they provides enough details for the purpose of replication or adaptation elsewhere. Please note that species scientific names should be italic. Please check ‘will be’. ‘water’ do you mean moisture content?. The use of ‘are’ or ‘were’ in the methodology. Please check the capital letters of some terms. Please justify or rephrase the sentence in page 6 ‘we can think that the atmospheric interference of this set of reflectance data can be ignored by…’.
The paper includes a section for accuracy assessment, which is good in such studies. The results are well presented in figures and tables and discussed with relevant to other relevant studies.
The discussion provides and nice synthesis for the use of remote sensing in ecological assessment of mangroves. However, the discussion relevant explicitly to the present study starts at the last two paragraphs ‘the research reveals that …’. It will be good to include more discussions related to the applications of the findings of the study in the local and regional contexts. For instance, even thought it is assumed that the three regions (A,B and C) have different degrees of human impacts, yet mangroves in these regions seem healthy.
Conclusion:
Conclusion is concise and reflect the study. Please rephrase ‘our research has improved our understanding’ to our research can improve..’.
